# Sustainable FinTech Innovation Orientation: A Moderated Model

**Manaf Al-Okaily** [1,*], **Abdul Rahman Al Natour** [2], **Farah Shishan** [3], **Ahmed Al-Dmour** [4], **Rasha Alghazzawi** [5] and **Malek Alsharairi** [6]

1   School of Business, Jadara University, Irbid 733, Jordan
2   Faculty of Administrative & Financial Sciences, University of Petra, Amman 11196, Jordan; abdulrahman.alnatour@up.edu.jo
3   School of Business, The University of Jordan, Amman 11942, Jordan; f.shishen@ju.edu.jo
4   School of Business, Al-Ahliyya Amman University, Amman 19328, Jordan; 102007@ammanu.edu.jo
5   King Talal School of Business Technology, Princess Sumaya University for Technology, Amman 1438, Jordan; r.alghzawi@psut.edu.jo
6   School of Management and Logistic Sciences, German Jordanian University, Amman 35247, Jordan; malek.alshareairi@gju.edu.jo
*   Correspondence: m.alokaily@jadara.edu.jo or m.oqaili@ju.edu.jo

**Abstract:** Financial technology (otherwise known as FinTech) refers to a type of technology and innovation that tries to improve and automate the delivery and use of financial services. Despite the importance of this technology in people's financial transactions in improving the management of their financial operations, processes, and lives, there is a lack of empirical evidence about sustainable FinTech services in the Jordanian context. Consequently, this research examines the factors that influence the acceptance of FinTech services, which have a variety of social, environmental, and ecological benefits. This study proposes an integrated model by combining the extended technology acceptance model (TAM) with the perceived enjoyment as an independent variable and electronic word of mouth (eWOM) as a moderator variable simultaneously. A total of 304 responses from Jordanian citizens were analyzed by the quantitative method of partial least squares structural equation modelling (PLS-SEM). The result confirmed that perceived usefulness and perceived enjoyment have a significant and positive influence on users' decision to use FinTech services. Meanwhile, eWOM is found to moderate the relationship between perceived usefulness and Jordanians' decisions to use FinTech services. Finally, this study provides practical implications for managers to encourage them to provide adequate, reliable, and sustainable services to their customers at a reasonable cost that fit their demands and ultimately improve their living standards. Current study limitations and future research directions are presented in the last section.

**Keywords:** sustainable FinTech; financial services; FinTech innovation; technology acceptance; financial inclusion; Jordanian context

## 1. Introduction

FinTech is widely regarded as one of the most critical developments in the financial business. It is a cutting-edge technology that aims to use computer programs and information technology to run traditional financial services [1]. It is rapidly evolving due to the sharing and circular economy, favorable regulation, and information technology [2]. FinTech has the potential to disrupt and reshape the financial industry by lowering costs, enhancing the quality of financial services, increasing employment rates, expanding financial access for businesses, reducing poverty by lowering transaction costs, and creating a more diverse and stable financial landscape [3,4]. In other words, FinTech is defined as "a new financial industry that applies technology to improve financial activities" [5] (p. 1).

Despite great expectations for FinTech's growth, it has not met those predictions in the actual world [6]. Customers are still cautious about accepting and using FinTech since it is creative yet inherently unpredictable, which has a negative impact on its growth. An

in-depth analysis of users' intention to adopt FinTech services is essential to achieve the sustainable development and growth of FinTech. FinTech services offer numerous benefits to users as well as the overall corporate network. End users may benefit from these services because they can save time, effort, and money [3]. An example of how FinTech services could help with food security and main sustainability challenges across the entire company network is as follows. FinTech-enabled digital markets could help agriculture's business process to be more sustainable by enhancing finance (e.g., crowdfunding) and distribution (e.g., digital payment systems). Farmers, landowners, investors, and consumers can all be connected to a digital network that fosters transparency, empowerment, resourcefulness, and public participation [4].

Aside from the benefits it provides to users and businesses by making operations and transactions considerably easier, FinTech acts as a catalyst for the long-term development of developing economies. FinTech services can help limit environmental effects by increasing productivity, efficiency, and cost savings; reducing product waste, chemicals, and resources; and measuring, analyzing, and tracking progress [7,8]. Sustainability has evolved from a specialist business concern to a mainstream one. FinTech can operate as a stimulus for collaborative innovation across traditional financial and banking organizations if it is well established. The sustainable development goals provide a common ground for businesses and stakeholders to develop long-term development [9]. FinTech has the potential to speed up the development of green and inclusive financial markets and realign money to promote long-term growth.

Lately, in several developing countries, such as Jordan, interest in improving the infrastructure for financial inclusion has risen, particularly in the FinTech services field to facilitate access to formal financial services for those who could not reach formal financial services. Despite the big commitment from these countries to deepen the financial inclusion of their unbanked citizens through FinTech services, FinTech adoption and use among financial users in the real world has not yet reached the predicted levels. As a result, it is critical to comprehend the primary elements influencing users' decisions to utilize FinTech services. Using an enhanced TAM, the current study will look into Jordanian citizens' decisions to adopt FinTech services.

Accordingly, this research focuses on two primary issues: Is there a direct relationship between the predictors (perceived usefulness, perceived enjoyment, and ease of use) and Jordanian citizens' decisions to use FinTech services? Does eWOM moderate the relationship between perceived usefulness, perceived enjoyment, and ease of use and Jordanian citizens' decisions to use FinTech services? Based on the findings, this study offers practical advice to service providers on the factors that influence Jordanian citizens' decisions to use FinTech services, which influence the industry's growth.

The current study proceeds as follows. Section 2 provides an overview of the research model and hypothesis development. Section 3 offers a detailed description of the methodology employed. Section 4 presents the research findings. Section 5 links the research findings with the extant literature and discusses how this study fulfilled the research objectives and implications. Section 6 presents future research directions and the study limitations. Finally, Section 7 presents an overall summary of this study.

## 2. Research Model and Hypotheses Development

The technology acceptance model (TAM) was first introduced by Davis [10] in the acceptance of the information technology (IT) domain. TAM is primarily interested in how users accept and use technology [11]. User acceptance is critical in the success of IT adoption [12,13]. Four steps are conducted to investigate the primary factors influencing users' intention to use a specific IT. The first step assesses how external factors affect the perceived usefulness and ease of use of IT. The degree to which users believe that adopting a particular innovation would improve their performance raises perceived usefulness. On the other hand, perceived ease of use refers to how confident users are that adopting a specific invention will require no physical effort. The second step occurs when users'

attitude toward using a particular innovation is influenced by its perceived usefulness and perceived ease of use. In the third step, perceived usefulness and attitude toward using a specific IT defines the usage intention. The final step is to decide whether or not to use a particular innovation [14].

Further, Venkatesh, Thong, and Xu. [15] recommended more validation of the information technology models in various contexts. Based on their recommendation, we extended TAM by including two additional constructs: perceived enjoyment and electronic word of mouth (eWOM). In total, four constructs allow for the analysis and assessment of their influence on Jordanian citizens' decisions to use FinTech services when compared with the original TAM. All the constructs mentioned above act as key direct and indirect predictors for digital financial inclusion usage. The relationships between the constructs are hypothesized based on a thorough literature analysis, which explains their combination into a concise and understandable research model, as represented in the grey-shaded parts in Figure 1.

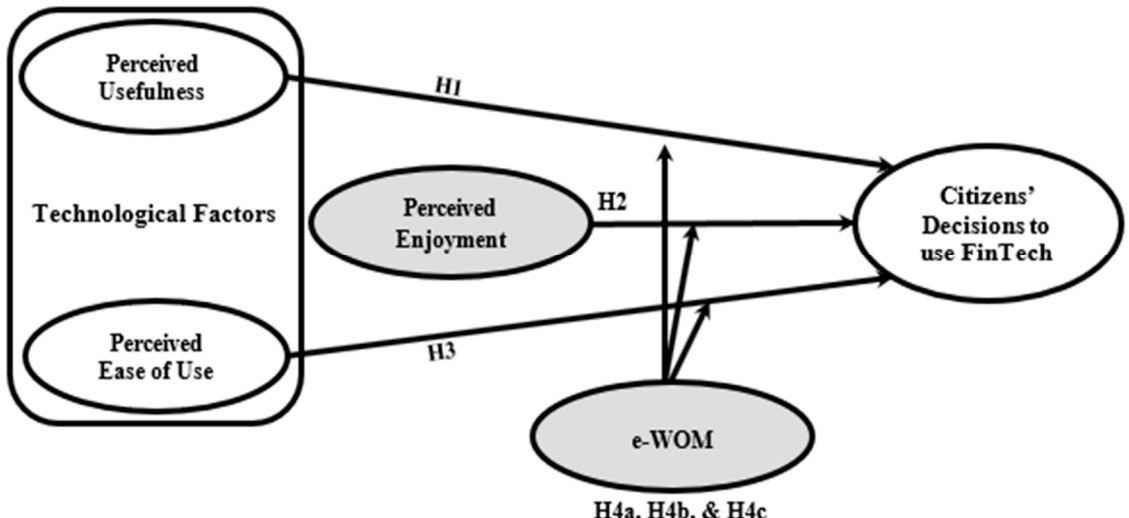

**Figure 1.** Research model.

### 2.1. Perceived Usefulness

According to Davis [10] (p. 320), perceived usefulness is defined as "the degree to which a person believes that using a particular system would enhance his or her job performance". Perceived usefulness is equivalent to performance expectancy in the unified theory of acceptance and use of technology (UTAUT) model, which is the degree to which people believe that employing proper technology will assist them to improve their job performance [16]. According to earlier research on the acceptance of digital financial system services, performance expectancy has a substantial and positive relationship with the intention to utilize these services [3,17]. As a result, we believe that if Jordanians perceive FinTech services to be beneficial and favorable, they will be more inclined to employ them in the future, and this puts forward the following hypothesis:

**Hypothesis 1 (H1).** *Perceived usefulness has a significant positive influence on Jordanian citizens' decisions to use FinTech services.*

### 2.2. Perceived Enjoyment

Perceived enjoyment is defined as the delight or pleasure obtained from utilizing technology, and it is equivalent to hedonic motivation in the UTAUT [18]. Perceived enjoyment has a more substantial influence on pleasure-oriented information systems than perceived usefulness, which indicates a higher utilitarian value [19]. For instance, when users wish to utilize the Internet during their coffee breaks or be occupied during their

leisure time, they may prefer m-channels over fixed channels [20]. As a result, earlier research in the fields of information systems and mobile technologies has confirmed the vital role of perceived enjoyment in this regard. Accordingly, it is worth mentioning that perceived convenience and enjoyment are two interconnected features of e-commerce [21].

According to Oyelami, Adebiyi, and Adekunle [22], there is a positive association between convenience and e-payment adoption in Nigeria. In addition, Pandey and Chawla [23] found that perceived enjoyment has a positive impact on e-commerce adoption among Indians. As a result of the perceived enjoyment associated with such systems, people may readily adopt them. In another study conducted in Saudi Arabia, Alalwan et al. [24] discovered that perceived enjoyment has a role in predicting Saudi customers' intention to adopt the mobile Internet. Based on the prior discussion, it is expected that the usage of FinTech services is perceived to be enjoyable, pleasant, and interesting among Jordanian citizens. Therefore, the following hypothesis is proposed:

**Hypothesis 2 (H2).** *Perceived enjoyment has a significant positive influence on Jordanian citizens' decisions to use FinTech services.*

### 2.3. Perceived Ease of Use

According to Davis [10] (p. 320), perceived ease of use is defined as "the extent to which users believe that applying a specific system would be free of efforts". More specifically, perceived ease of use is equivalent to effort expectancy in the UTAUT, which refers to the amount of work required to utilize any system regardless of how simple it is [16]. According to prior research in the field, perceived ease of use is a critical component in the acceptance of information systems [17,25]. Accordingly, it is believed that if people's perceptions of using FinTech services are effortless, they will be more likely to utilize them. Therefore, the following hypothesis is suggested:

**Hypothesis 3 (H3).** *Perceived ease of use has a significant positive influence on Jordanian citizens' decisions to use FinTech services.*

### 2.4. Electronic Word of Mouth

Electronic word of mouth (eWOM) communication can be viewed as a positive or negative statement about a product or company or new technologies through the Internet that is available to a group of people and institutions through the mobile Internet and device [26]. Consumers can use online discussion and forums, social networking sites, and consumer review sites to communicate and exchange information about a particular product or service through eWOM [27]. eWOM has the potential to influence the uptake of online banking services. Several scholars found that IT adoption can be influenced directly and indirectly by eWOM through influencing attitudes [28–30], as well as buying intention (e.g., Bebber et al. [28] and Vahdati and Nejad [31]). Further, when consumers find the information beneficial in the context of making a purchasing decision, they are expected to embrace eWOM.

Perceived usefulness, ease of use, attitude toward behavior are associated with eWOM [32]. Forcedly, these factors can drive IT adoption through behavioral intention and purchasing behavior. According to Luarn and Lin [33], several researchers have tested and validated the adoption of FinTech services, such as Leiva et al. [34] and Sharma et al. [35]. Moreover, various factors linked to IT adoption have been studied, such as trust (e.g., Cabanillas et al. [36]) and perceived costs (e.g., Bhatiasevi [37]). Consequently, an integrated model, including eWOM, highly cited IT adoption TAM, and other related concepts, is suitable as a theoretical foundation. In our framework, consumers perceive eWOM to be useful when the information received leads to the acceptance or rejection of FinTech services. According to the previous literature review, we hypothesize the following:

**Hypothesis 4a (H4a).** *eWOM moderates the relationship between perceived usefulness and Jordanian citizens' decisions to use FinTech services.*

**Hypothesis 4b (H4b).** *eWOM moderates the relationship between perceived enjoyment and Jordanian citizens' decisions to use FinTech services.*

**Hypothesis 4c (H4c).** *eWOM moderates the relationship between perceived ease of use and Jordanian citizens' decisions to use FinTech services.*

To sum up, the earlier sections discussed the direct relationships between perceived usefulness, perceived enjoyment, and perceived ease of use as the independent variables and Jordanian citizens' decisions to use FinTech services as the dependent variable. Then, the indirect relationship between perceived usefulness, perceived enjoyment, and perceived ease of use and Jordanian citizens' decisions to use FinTech services moderated by eWOM was discussed. Accordingly, six hypotheses were formulated to test the various relationships in an integrated research model to predict the determinants and opportunities influencing Jordanian citizens' decisions to use FinTech services.

### 3. Research Methodology

A self-administered survey was used in this study to collect the desired data to test the proposed model of the current research. Researchers conducted a survey with a purposive sample size of 438 questionnaires among Jordanian citizens to get the required data from various universities and ministries of Jordan. A total of 352 questionnaires were returned, giving an 80% response rate, with 48 being excluded. However, only 304 responses were usable for further analysis; the effective response rate was approximately 69%. Of the 304 respondents, 161 were male, and 143 were female. The ages of the respondents indicated that the majority of them were less than 30 years old, representing 149 responses. In contrast, the respondents were aged between 41 and 50 years old, between 30 and 40 years old, and more than 50 years old, representing 73, 65, and 17 individuals, respectively.

Furthermore, concerning devices owned by the respondents, a high percentage of the respondents in this study had smartphones, representing 287 of the total responses, while 17 had only regular phones. Finally, concerning the respondents' place of residence, most of them in this study live in the central region, representing 176 responses. On the other hand, other respondents live in the northern and southern regions, representing 77 and 51 individuals, respectively (see Table 1).

**Table 1.** Demographic characteristics of the respondents.

| Category | Coding | Frequency |
|---|---|---|
| Gender | Male | 161 |
| | Female | 143 |
| | Total | 304 |
| Age | Less than 30 years | 149 |
| | 30–40 years | 65 |
| | 41–50 years | 73 |
| | More than 50 years | 17 |
| | Total | 304 |
| Type of mobile device | Normal phones | 17 |
| | Smartphones | 287 |
| | Total | 304 |
| Place of residence | Northern region | 77 |
| | Central region | 176 |
| | Southern region | 51 |
| | Total | 304 |

Jordan was chosen as the study's focus because it is at the forefront of technological innovation, providing tangible support to institutions and technology businesses alike and attempting to establish a regulatory framework that will assist in technology drive

and build Jordan's economy [38]. Jordan is one of the Middle East's developing countries. Furthermore, the Jordanian Telecommunications Regulatory Commission [39] conducted some statistical analysis on the penetration of mobile devices in general and discovered that the penetration rate of smart mobile phones in Jordan is up to 93.3%, which is a high proportion. Moreover, 88% of families have access to the Internet at home. Furthermore, with a mobile Internet penetration rate of 8.8 million users, Jordan's market acceptance of technology is relatively strong, making it an appropriate sample for analyzing their decisions to use FinTech services.

As part of the data collection process, the researchers used several tactics to reach the intended respondents. Participants were gathered from all over the country. The researchers collected data from employees working in various ministries and public sector institutions, which may explain the high number of answers from the central area. Most of these institutions are in Amman, the capital of Jordan, which is at the heart of the country. Further, students from three of Jordan's most prestigious public universities, Mutah University, Yarmouk University, and the University of Jordan, were also targeted. These universities were chosen because of their large student population size and geographical location. They are located in Jordan's southern, northern, and central areas, which helps generalize the predicted results by covering Jordan's three geographical regions. Mainly, the researchers visited each university and distributed the questionnaires to lecturers and students after receiving permission to gather data from the president's office. In addition, the questionnaires were distributed to students in classrooms with the cooperation of their lecturers. This method provided an excellent opportunity to meet with a large number of students under the supervision of their lecturers, which encouraged them to answer the questionnaires.

In conclusion, a 7-point Likert scale was used in this study to measure the responses since this scale has been widely used in social science studies [40]. In this regard, researchers indicate that a 7-point scale is just as good as other scales because this scale provides a wider variance among the measures and offers a much more comprehensive range of options [41]. In addition, Foddy and Foddy [42] concluded that a minimum 7-point scale is required to ensure scale reliability and validity. It was necessary to revisit some potential participants to increase the response rate (e.g., students at their campus). At this stage, all the participants were given a brief about the study and its importance. However, some challenges were faced. For example, some students refused to spend their time answering the questions, and others refused to answer the questions because they did not know what FinTech services are. Nonetheless, there was an increase in the number of people who responded.

## 4. Research Results

To achieve the purposes of this study, PLS software was utilized to assess the proposed research model. PLS avoids minor sample size concerns and is less strict in the expectations of normality distribution and error term requirements [43]. Besides, PLS can test both measurement and structural models simultaneously [44,45]. In addition, PLS can work with multifaceted models with a hierarchical structure and many components, indicators, and relationships [46]. Similarly, PLS is capable of modeling higher-order constructs [47]. PLS also allows for a more flexible treatment of advanced model elements, such as mediating and moderating factors [48].

### 4.1. PLS Outer Model

According to Hair et al. [49], the outer (measurement) model assessment is the first prerequisite step for generating outcomes in PLS by analyzing the measurements' reliability and validity. The results displayed in Table 2 confirm that all traditional standards of validity and reliability are within the acceptance criteria. Accordingly, the current research can safely progress toward inner (structural) model analysis and examining the proposed research hypotheses.

**Table 2.** Factor loading, Cronbach's alpha, CR, AVE, and HTMT.

| Latent Variable | Item Code | Reliability | | | Validity | |
|---|---|---|---|---|---|---|
| | | **Indicator Reliability** | **Internal Consistency Reliability** | | **Convergent Validity** | **Discriminant Validity** |
| | | Factor Loadings | Cronbach's Alpha | CR | AVE | HTMT |
| | | Loading >0.70 or >0.40 and has no impact on AVE and CR | $\alpha \geq 0.70$ | $CR \geq 0.70$ | $AVE \geq 0.50$ | $HTMT < 0.90$ |
| **Perceived Usefulness** | PU1 | 0.954 | 0.942 | 0.962 | 0.895 | Suitable |
| | PU2 | Dropped | | | | |
| | PU3 | 0.945 | | | | |
| | PU4 | 0.940 | | | | |
| **Perceived Enjoyment** | PE1 | 0.953 | 0.901 | 0.917 | 0.813 | Suitable |
| | PE2 | 0.921 | | | | |
| | PE3 | 0.932 | | | | |
| **Perceived Ease of Use** | PEU1 | Dropped | 0.921 | 0.944 | 0.849 | Suitable |
| | PEU2 | 0.926 | | | | |
| | PEU3 | 0.918 | | | | |
| | PEU4 | 0.920 | | | | |
| **Electronic Word of Mouth** | eWOM1 | 0.901 | 0.892 | 0.921 | 0.823 | Suitable |
| | eWOM2 | 0.955 | | | | |
| | eWOM3 | 0.892 | | | | |
| | eWOM4 | 0.877 | | | | |
| **Decisions to Use FinTech** | DUT1 | 0.927 | 0.957 | 0.969 | 0.885 | Suitable |
| | DUT2 | 0.957 | | | | |
| | DUT3 | 0.942 | | | | |
| | DUT4 | 0.936 | | | | |

*4.2. PLS Inner (Structural) Model*

The subsequent step in the PLS analysis after confirming that the outer (measurement) model fitted the accepted standards of reliability and validity is the assessment of the inner (structural) model and the examination of the proposed research hypotheses [49]. As a result, using the bootstrapping method to assess path coefficients entails the smallest bootstrap sample of 5000, and the number of cases should be equal to the number of observations in the original sample [46,48]. Along this vein, the critical values for a two-tailed test are 1.65 (SL: 10%), 1.96 (SL: 5%), and 2.57 (SL: 1%), which indicate that path coefficients with a probability of error of 5% or less are commonly considered significant [48]. Therefore, to construct standard errors and acquire *t*-statistics, the researchers used resampling of 5000 with a replacement number from the bootstrap cases equal to the original number of samples (304).

In line with test direct and indirect effect interrelationships, the primary direct effect model was examined without including the moderating role of eWOM. The moderation effect was examined in a pre-existing model known as the interaction model in the second step. In total, six hypotheses were declared to assess the proposed research hypotheses, as illustrated in Figure 1. As a result, all hypotheses were accepted except for hypotheses H3, H4b, and H4c, as depicted in Table 3.

**Table 3.** Results of hypotheses testing of the study.

| No. | Relationship | | | | Original Sample | Standard Deviation | T Statistics | P Statistics | Decision |
|---|---|---|---|---|---|---|---|---|---|
| | IV | | DV | | | | | | |
| H1 | | PU⟶DUT | | | 0.245 | 0.057 | 4.298 | 0.000 | Accepted |
| H2 | | PE⟶DUT | | | 0.192 | 0.078 | 2.461 | 0.010 | Accepted |
| H3 | | PEU⟶DUT | | | 0.006 | 0.021 | 0.295 | 0.768 | Rejected |
| **No.** | **IV** | **MOD** | **DV** | | | | | | |
| H4a | PU | eWOM | DUT | | 0.233 | 0.075 | 3.106 | 0.001 | Accepted |
| H4b | PE | eWOM | DUT | | 0.065 | 0.052 | 1.243 | 0.214 | Rejected |
| H4c | PEU | eWOM | DUT | | −0.062 | 0.093 | -0.663 | 0.507 | Rejected |

## 5. Discussions and Implications

The empirical findings confirmed that perceived usefulness positively and significantly influences the decisions to use FinTech services, and thus, hypothesis 1 was accepted. This finding is in line with the work completed by Davis [10] in TAM, which states that perceived usefulness increases from the extent to which users think embracing a specific innovation will improve their performance. This suggests that Jordanians are more likely to increase their adoption of FinTech services if they see the value of such systems. These technologies and services may be valuable because they save time, effort, and money while decreasing product waste, chemicals, and resources [7,8].

The results reveal a positive and significant association between perceived enjoyment and Jordanian citizens' decisions to utilize FinTech services, and thus, hypothesis 2 was accepted. This finding is in line with prior studies, such as that of Alalwan et al. [24], who claimed that perceived enjoyment plays a significant role in predicting Saudi customers' intention to adopt the mobile Internet. This leads us to suggest that managers should make the experience of using FinTech services more fun, enjoyable, and exciting, and focus on the entire customer experience and making it more engaging.

Surprisingly, no other relationships between perceived ease of use and decisions to use FinTech services could be found; hence, hypothesis 3 was rejected. Such results contradict the work of Davis [10] in TAM. This finding indicates that Jordanians believe that it is challenging to use FinTech services and that these innovations require more work and effort.

In terms of eWOM's moderating role, the *t*-value is positive and significant (t-value = 3.106, $p < 0.01$). This, in turn, supports the moderating role of eWOM in the relationship between perceived usefulness and decisions to use FinTech services. In other words, eWOM strengthens the positive relationship between perceived usefulness and decisions to use FinTech services. As shown in Figure 2, under a high level of eWOM, perceived usefulness is more predictive of decisions to use FinTech services. The findings demonstrate that managers must recognize the essential impact of eWOM on how others view the benefits of using FinTech services, which leads to more influence on others' decisions to use these services, resulting in more FinTech adoption. Users are particularly interested in reading both bad and positive reviews of other users' experiences. They may also encourage current users to post reviews and comments on various social media platforms, forums, and blogs. Users are particularly interested in reading both bad and positive reviews of other users' experiences. According to other studies, these positive and negative reviews have an impact on potential users' decisions [50–52]. It is worth noting that these reviews might be about the potential benefits of FinTech services, as well as people's beliefs about the necessity of establishing a sustainable society and environment, and how each of them should contribute to protecting the environment and the planet.

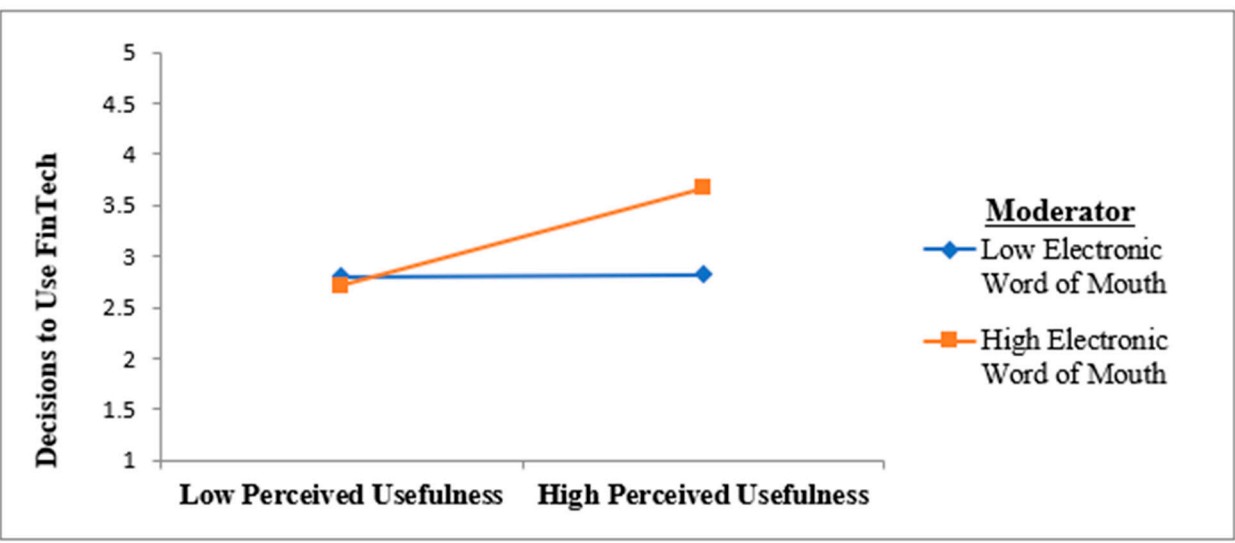

**Figure 2.** Interaction effect between perceived usefulness and decisions to use FinTech.

## 6. Limitations and Future Research Directions

This study is subject to a particular limitation as it is confined to the Jordanian context, which cannot be generalized to other countries; however, the study provides strategic guidance for policymakers in building a framework to promote FinTech adoption, which can foster financial inclusion. Thus, further research is needed in other locations, such as the Gulf Cooperation Council or Middle Eastern and North African regions, to strengthen the generalizability of these findings. Finally, the current research employed one of the common models in technology acceptance before the COVID-19 pandemic, namely, TAM, and combined only two external factors with it: perceived enjoyment and electronic word of mouth. Therefore, future research is required to obtain a deeper understanding of FinTech services' usage situation after the COVID-19 pandemic by proposing new important factors related to the pandemic context (e.g., fear of COVID-19 pandemic and its influence on FinTech services usage).

## 7. Conclusions

The unsatisfactory level of adoption of FinTech innovations in Jordan requires a particular research focus to understand why, how, and when the adoption of these innovations will take place. This study has increased our understanding of the topic by identifying several determinants that influence the adoption of FinTech services in an attempt to understand the conflicting conclusions regarding the impact of these determinants. This study aimed to answer the research questions set earlier. The theoretical model in this study is based on TAM. The key constructs in TAM (perceived usefulness and perceived ease of use) were utilized as immediate predictors of Jordanian citizens' decisions to use FinTech services. In addition, new constructs, namely, perceived enjoyment and eWOM, were incorporated as an extension for TAM. Therefore, the variables involved in the current study were grouped into six hypotheses formulated to answer the research questions. Finally, Jordanian citizens' decisions to use FinTech services are found to be significantly and positively influenced by perceived usefulness and perceived enjoyment. Furthermore, eWOM moderates the relationship between perceived usefulness and Jordanian citizens' decisions to use FinTech services.

**Author Contributions:** Conceptualization, A.R.A.N., R.A.; methodology, F.S.; software, M.A.-O.; validation, M.A.-O.; formal analysis, M.A.-O.; investigation, R.A.; resources, M.A.-O., data curation, M.A.-O.; writing—original draft preparation, A.R.A.N.; A.A.-D. writing—review and editing, M.A.; visualization, A.A.-D.; supervision, F.S.; project administration, M.A. All authors have read and agreed to the published version of the manuscript.

**Funding:** This research received no external funding.

**Institutional Review Board Statement:** Not Applicable.

**Informed Consent Statement:** Informed consent was obtained from all subjects involved in the study.

**Data Availability Statement:** Not Applicable.

**Acknowledgments:** We highly acknowledge the valuable insights of the reviewers and editor who have contributed extensively in improving the article's quality. The authors would also like to thank the participants in this research for their cooperation in providing the information that contributed to the success of this research.

**Conflicts of Interest:** The authors declare no conflict of interest.

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
