# Peer review of "Sustainable FinTech Innovation Orientation: A Moderated Model"

_sustainability, doi:10.3390/su132413591_

Round 1

Reviewer 1 Report

This is an interesting research paper that is supposed to focus on sustainable fintech innovation orientation.

However, apart from the introduction, which covers the importance of sustainable fintech, the focus of the analyses is limited to the acceptance of the fintech innovations in Jordan. The researcher used a reliable methodology (TAM Model) and carried out a survey on a reliable sample, but still, the research questions do not clearly explain the connection between sustainable finance. The author should explain why and how acceptance of the fintech applications can impact the sustainability of fintech innovation. 

Important references are also missing on the topic:

  • Ryu, H. S., & Ko, K. S. (2020). Sustainable development of Fintech: Focused on uncertainty and perceived quality issues. Sustainability12(18), 7669. 
  • Shin, Y. J., & Choi, Y. (2019). Feasibility of the FinTech industry as an innovation platform for sustainable economic growth in Korea. Sustainability11(19), 5351.
  • Macchiavello, E., & Siri, M. (2020). Sustainable Finance and Fintech: Can Technology Contribute to Achieving Environmental Goals? A Preliminary Assessment of ‘Green FinTech'.

In short, a good model and consistent analyses are carried out but the research question is confusing. The sustainability of fintech cannot be explained by the acceptance of fintech applications.

Author Response

Dear Respected Editors and Reviewers, 
Kindly find the revised version of my paper,  I inserted the list of comments in yellow colour as requested. I would like to take this opportunity to extend my appreciation to you and the reviewers for their insightful comments on the paper, as these comments led me to an improvement of the work. My revisions reflect all reviewers’ suggestions. Finally, really I am grateful for your assistance as well as your valuable guidance.

Thank you for your consideration of this manuscript.

Sincerely,
Corresponding author
Dr. Manaf Al-Okaily

Reviewer 2 Report

The researchers propose an integrated model by integrating the extended Technology Acceptance Model (TAM) with the perceived enjoyment as an independent variable and electronic word of mouth (eWOM) as moderator variable simultaneously. 

This paper discusses on the main research question “Can an extended TAM model be applied to determine the acceptance rate of FinTech services usage among the citizens in Jordan?”. The following questions are defined to support the main issues above - 

(1) Is there a direct relationship between the predictors (perceived usefulness, perceived enjoyment and perceived ease of use) and the behavioural intention to use FinTech services?

(2) Does the eWOM moderate the relationship between (perceived usefulness, perceived enjoyment and perceived ease of use) and the behavioural intention to use FinTech services. 

The research findings suggested hypotheses that behavioural intention to use FinTech services is significantly and positively influenced by perceived usefulness and perceived enjoyment.

The works are interesting in order to perceive the important of having the sustainable FinTech for the related stakeholders. Some observations to improve the paper are as follows:

Table 3. Results of hypotheses testing of the study  ==> This table need to properly the important of information to be presented.

Author Response

(The authors gave the same response as above.)

Reviewer 3 Report

Thank you for inviting me to review this manuscript, titled “Sustainable FinTech Innovation Orientation: A Moderation Perspective”. The aim of this study is to analyze the factors that influence FinTech services sustainability. The research topic proposed by the authors is relevant and is of current interest.

Please find my detailed comments below:

I suggest a critical approach to the relevant scientific papers in the field and emphasis the aspects which would have supported the authors' contribution to the existing debates in the academic literature. The authors declare the aim without underlining the studies’ gap.

If it is possible to expand the methodology section a bit more, indicating the type of design of the questionnaire, the representativeness, etc.

Some aspects can be included in the debate. Reference to sustainability is made in introduction, while in the empirical study and conclusions no specific aspects are considered; the attempt to link the study with sustainability is merely superficial. Is there a link between Financial Inclusion, that depends on FinTech, and Sustainability?  

No reference is made to whether or not the results are in line with previous research. I suggest that the conclusions should be concreted by associating with previous studies in the literature.

Please take into account the Research Manuscript Sections of the journal and do a proof reading.

Kind Regards

Author Response

(The authors gave the same response as above.)

Round 2

Reviewer 1 Report

The author(s) addressed most of the previous concerns.

Reviewer 3 Report

The paper has been improved.

Kind regards